# Female sex protects against renal edema, but not lung edema, in mice with partial deletion of the endothelial barrier regulator Tie2 compared to male sex

Anoek L. I. van Leeuwen[1,2], Elise Beijer[1,3,4], Roselique Ibelings[1,4], Nicole A. M. Dekker[1], Marjolein R. A. van der Steen[1], Joris J. T. H. Roelofs[5], Matijs van Meurs[6,7], Grietje Molema[7], Charissa E. van den Brom[1,4,8]*

1 Department of Anesthesiology, Amsterdam UMC, VU University, Amsterdam, The Netherlands, 2 Department of Physiology, Amsterdam UMC, VU University, Amsterdam, The Netherlands, 3 Department of Surgery, Amsterdam UMC, VU University, Amsterdam, The Netherlands, 4 Laboratory of Experimental Intensive Care and Anesthesiology, Amsterdam UMC, University of Amsterdam, Amsterdam, The Netherlands, 5 Department of Pathology, Amsterdam UMC, University of Amsterdam, Amsterdam, The Netherlands, 6 Department of Critical Care, University Medical Center Groningen, Groningen, the Netherlands, 7 Department of Pathology and Medical Biology, University Medical Center Groningen, Groningen, The Netherlands, 8 Department of Intensive Care Medicine, Amsterdam UMC, University of Amsterdam, Amsterdam, The Netherlands

* c.vandenbrom@amsterdamumc.nl

## Abstract

### Background

The endothelial angiopoietin/Tie2 system is an important regulator of endothelial permeability and targeting Tie2 reduces hemorrhagic shock-induced organ edema in males. However, sexual dimorphism of the endothelium has not been taken into account. This study investigated whether there are sex-related differences in the endothelial angiopoietin/Tie2 system and edema formation.

### Methods

Adult male and female heterozygous Tie2 knockout mice (Tie2$^{+/-}$) and wild-type controls (Tie2$^{+/+}$) were included (n = 9 per group). Renal and pulmonary injury were determined by wet/dry weight ratio and H&E staining of tissue sections. Protein levels were studied in plasma by ELISA and pulmonary and renal mRNA expression levels by RT-qPCR.

### Results

In Tie2$^{+/+}$ mice, females had higher circulating angiopoietin-2 (138%, p<0.05) compared to males. Gene expression of angiopoietin-1 (204%, p<0.01), angiopoietin-2 (542%, p<0.001) were higher in females compared to males in kidneys, but not in lungs. Gene expression of Tie2, Tie1 and VE-PTP were similar between males and females in both organs. Renal and pulmonary wet/dry weight ratio did not differ between Tie2$^{+/+}$ females and males. Tie2$^{+/+}$

**Data Availability Statement:** All data have been made available via: Sex differences and loss of Tie2

- EASY (knaw.nl) https://easy.dans.knaw.nl/ui/datasets/id/easy-dataset:315203/tab/2.

**Funding:** This research was supported by the Dutch Heart Foundation (2016T064, to NAMD, https://www.hartstichting.nl/), Dutch Research Council (ZonMW, Veni Grant 2019, to CEvdB, https://www.zonmw.nl/nl), European Society of Intensive Care Medicine (Basic research Award 2021 to CEvdB, https://www.esicm.org/) and European Society of Anaesthesiology and Intensive Care (ESAIC Research project grant 2022 to CEvdB, https://www.esaic.org/). The funders had no role in study design, data collection and analysis, decision to publish, or preparation of the manuscript.

**Competing interests:** The authors have declared that no competing interests exist.

females had lower circulating NGAL (41%, $p < 0.01$) compared to males, whereas renal NGAL and KIM1 gene expression was unaffected.

Interestingly, male Tie2$^{+/-}$ mice had 28% higher renal wet/dry weight ratio ($p < 0.05$) compared to Tie2$^{+/+}$ males, which was not observed in females nor in lungs. Partial deletion of Tie2 did not affect circulating angiopoietin-1 or angiopoietin-2, but soluble Tie2 was 44% and 53% lower in males and females, respectively, compared to Tie2$^{+/+}$ mice of the same sex. Renal and pulmonary gene expression of angiopoietin-1, angiopoietin-2, estrogen receptors and other endothelial barrier regulators was comparable between Tie2$^{+/-}$ and Tie2$^{+/+}$ mice in both sexes.

## Conclusion

Female sex seems to protect against renal, but not pulmonary edema in heterozygous Tie2 knock-out mice. This could not be explained by sex dimorphism in the endothelial angiopoietin/Tie2 system.

## Introduction

Sex-related differences in outcome of traumatic hemorrhagic shock (THS) patients gained more attention since 2010. Although THS remains associated with high mortality, several studies have demonstrated a protective effect of female gender in response to THS [1–5]. Female gender is associated with a 2% decreased risk of acute kidney injury [2–5], a 9% decreased risk of lung injury [5] and a 14–21% decreased mortality rate in patients following THS [2, 6–8]. To date, sex-related differences in outcome following THS are unexplained and understanding the differences between male and female THS patients could contribute to the development of novel treatment strategies.

THS causes a systemic inflammatory response that activates the endothelium [9]. This results in increased endothelial permeability, leakage of fluids into the interstitium and tissue edema [10]. We have previously shown that plasma from male THS patients induces endothelial hyperpermeability *in vitro* [11] and that THS in male rats induces vascular leakage in both lung and kidney [12]. One of the molecular systems involved in regulation of endothelial permeability is the endothelial angiopoietin/Tie2 system (Fig 1).

The activation of the endothelial receptor Tie2 is tightly controlled by the interplay between the paralogous endothelial receptor Tie1, the tyrosine phosphatase vascular endothelial protein tyrosine phosphatase (VE-PTP), and two secreted ligands, angiopoietin-1 and angiopoietin-2 [13]. The latter can act as agonist, partial agonist, or antagonist depending upon context [14]. In healthy conditions, angiopoietin-1 phosphorylates and thereby activates the endothelial Tie2 receptor, which assists in maintaining endothelial barrier function via inhibition of RhoA kinase and activation of small GTPase Rac-1 [15, 16]. During stress, circulating angiopoietin-2 and the expression of the VE-PTP receptor are increased. Both can inhibit phosphorylation and thereby prevent activation of Tie2. This consequently disrupts the endothelial barrier with vascular leakage and edema as a result [15, 17]. Additionally, Tie2 signaling and endothelial barrier function can be affected by other receptors, such as integrin α5β1, and vascular endothelial growth factor (VEGF). VEGF can increase permeability via inhibition of vascular endothelial (VE)-cadherin adherens junctions [18], and is a known promotor of shedding of the Tie2 receptor [19]. On the other hand integrin α5β1 promotes Tie2 phosphorylation when angiopoietin-1 levels are low [20].

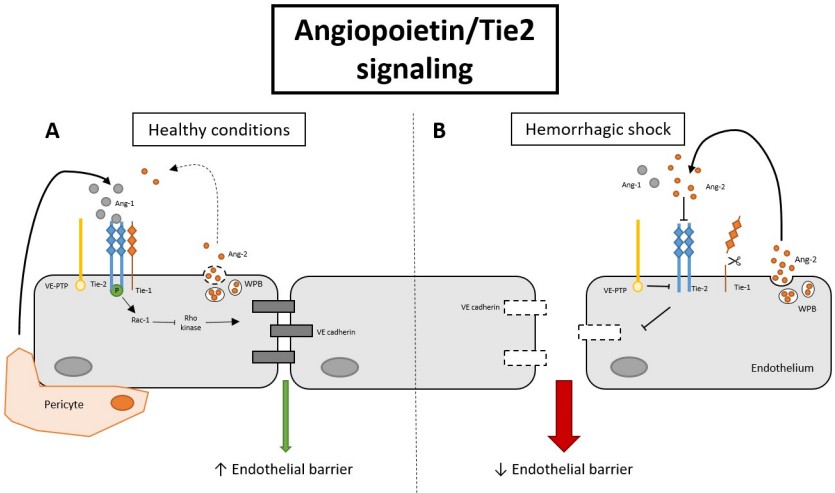

**Fig 1. Schematic overview of the effect of angiopoietin/Tie2 signaling on endothelial barrier function.** In quiescence (**A**), angiopoietin-1 (ang-1) is released from pericytes and activates, and thereby phosphorylates tyrosine kinase receptor Tie2. Activation of Tie2 strengthens endothelial barrier function via Rac-1/Rho kinase/vascular endothelial (VE)-cadherin signaling. In contrast, in response to hemorrhagic shock (**B**) angiopoietin-2 (ang-2) is rapidly released from weibel palade bodies (WPB), leading to increased endothelial permeability via inhibition of Tie2 activation. Other transmembrane proteins that affect Tie2 phosphorylation include Tie1 and vascular endothelial-protein tyrosine phosphatase (VE-PTP), which both inhibit Tie2 phosphorylation upon a stress stimulus.

We and others have previously shown that THS disturbs endothelial angiopoietin/Tie2 signaling, including reduced expression of the endothelial Tie2 receptor in the kidney and lungs [12, 21] and increased circulating levels of angiopoietin-2 [12]. Interestingly, we have been able to reduce vascular leakage by pharmacologically activating endothelial Tie2 in male THS rats [12], suggesting that targeting Tie2 is a promising strategy to reduce organ injury [22, 23]. Unfortunately, sexual dimorphism of the endothelium in THS has not been taken into account [24]. Here, we investigated the effect of sex on renal and pulmonary edema in heterozygous mice with genetically reduced endothelial Tie2 expression thereby mimicking the suppressive effect of THS on expression of the endothelial Tie2 receptor. In addition, we investigated whether expression of components of the endothelial angiopoietin/Tie2 system in kidneys and lungs differs between healthy female and male mice.

## Materials and methods

### Study approval

Analyses were performed on tissue samples of terminated animals from breeding surplus provided by the Amsterdam Animal Research Center of the VU University Amsterdam, the Netherlands. These experiments do not require an animal research permit from the Central Committee for Animal Experiments (Centrale Commissie Dierproeven) in the Netherlands. These experiments make an important contribution to the *replacement* and *reduction* principles in animal experimentation according to animal welfare regulations in full agreement with the Directive 2010/63/EU. All procedures were conducted following the European Convention for the Protection of Vertebrate Animals used for Experimental and Other Scientific Purposes and the ARRIVE 2.0 guidelines on animal research [25]. Euthanasia was performed under isoflurane anesthesia, and all efforts were made to minimize suffering.

## Animals and genotyping

Heterozygous Tie2 knockout mice (exon 9 deletion, C57BL6/J background) were generated by Jongman *et al.* and bred as previously described [26]. In the current study, heterozygous Tie2 knockout mice (Tie2$^{+/-}$) with partially reduced Tie2 expression and wild-type littermate controls (Tie2$^{+/+}$) with normal Tie2 expression were included.

Mouse genomic DNA was extracted from ear punches using standard protocols. The genotype of Tie2$^{+/-}$ mice was determined by PCR using the primers 5′-GGGCTGCTACAATAGCTT TGG-3′ and 5′-GTTATGTCCAGTGTCAATCAC-3′ resulting in a 644 bp PCR product when exon 9 is still present (Tie2$^{+/+}$) and in a 309 bp PCR product when exon 9 of Tie2 was excised by Cre-recombinase (Tie2$^{+/-}$). PCR products were run on a 1.5% (w/v) agarose gel in Tris-borate-EDTA-buffer with 0.005% (v/v) ethidium bromide, and visualized under UV light (S1 Fig).

## Experimental set-up

Male and female mature adult mice from breeding surplus (bred in the animal facility of the VU University, Amsterdam, the Netherlands, generation N2/F3 and N2/F4) were included at the age of 3 to 6 months. Mature mice of at least 3 months old are considered sexual mature and represent humans between 20 and 30 years of age [27]. Mice were housed in a temperature-controlled room (12/12 h day/night cycle, 20–23 ˚C, 40–60% humidity) with food (Teklad global 18% protein, catalog nr 2918CS, Envigo, Indianapolis, USA) and autoclaved tap-water *ad libitum*. Mice were terminated via decapitation after sedation with 5% isoflurane in air at the animal facility. For both sexes, Tie2$^{+/+}$ and Tie2$^{+/-}$ mice were included (n = 9 per group). Upon sacrifice, the right kidney and lung were used to determine edema formation by wet/dry weight ratio. The left kidney and lung were snap frozen in liquid nitrogen and stored at -80˚C for further analyses. Whole blood was collected in heparin tubes, centrifuged twice at 4˚C (10 min at 4.000×$g$ and 15 min at 10.000×$g$) to obtain platelet free plasma and stored at -80˚C. The investigators (AvL, NAMD, MRAvdS, JJTHR) performing the *ex vivo* analyses were blinded to group allocation.

## Renal and pulmonary edema formation

Renal and pulmonary tissue was harvested immediately upon sacrifice. Wet tissue was weighed and subsequently dried at 70˚C. After 24 hours, dry tissue was weighed and wet/dry weight ratio was calculated as estimate of tissue water content.

## Plasma analyses

Levels of circulating angiopoietin-1 (LS-F2956, LSBio, Seattle, USA), angiopoietin-2 (MANG20, R&D systems, Minneapolis, MN, USA), soluble Tie2 (MTE200, R&D systems, Minneapolis, MN, USA), and neutrophil gelatinase-associated lipocalin (NGAL; KIT042, Bioporto, Hellerup, Denmark) were measured with Enzyme-linked immunosorbent assay (ELISA) in accordance to the manufacturer.

## RNA analyses

Total RNA was extracted from 10–20 mg frozen whole kidney and lung tissue and isolated using the RNeasy mini kit (Qiagen, Venlo, the Netherlands) as previously described [12]. RNA concentration and purity were determined using NanoDrop 1000 (NanoDrop Technologies, Wilmington, DE, USA). A total of 1 μg RNA was transcribed into complementary DNA using an iScript cDNA synthesis kit (Bio-Rad, Veenendaal, the Netherlands) using oligo-dT priming.

**Table 1. Real time-qPCR primers.**

| Gene | Assay ID | Encoded protein |
|---|---|---|
| Gapdh | Mm99999915_g1 | Glyceraldehyde-3-phosphate dehydrogenase (GAPDH) |
| Angpt1 | Mm00456503_m1 | Angiopoietin-1 |
| Angpt2 | Mm00545822_m1 | Angiopoietin-2 |
| Tek | Mm00443243_m1 | Endothelial-specific receptor tyrosine kinase (Tie2) |
| Tie1 | Mm00441786_m1 | Tyrosine kinase with immunoglobulin-like and EGF-like domains 1 (Tie1) |
| Ptprb | Mm00459467_m1 | Protein tyrosine phosphatase, receptor type, B (VE-PTP) |
| Esrra | Mm00433143_m1 | Estrogen related receptor, alpha |
| Esrrb | Mm00442411_m1 | Estrogen related receptor, beta |
| Itga5 | Mm00439797_m1 | Integrin alpha 5 (fibronectin receptor alpha) |
| Itgb1 | Mm01253230_m1 | Integrin beta 1 (fibronectin receptor beta) |
| Rac1 | Mm01201653_mH | RAS-related C3 botulinum substrate 1 |
| RhoA | Mm00834507_g1 | Ras homolog gene family, member A |
| Vegfa | Mm00437306_m1 | Vascular endothelial growth factor A |
| Havcr1 | Mm00506686_m1 | Hepatitis A virus cellular receptor 1 (Havcr1), Kidney injury marker 1 (KIM-1) |
| Lcn2 | Mm01324470_m1 | Lipocalin 2 (NGAL) |

mRNA abundance was measured using a CFX384 Touch real-time PCR detection system (Bio-Rad, Veenendaal, the Netherlands). mRNA abundance of angiopoietin-1, angiopoietin-2, Tie2, Tie1, vascular endothelial protein tyrosine phosphatase (VE-PTP), estrogen receptor α, estrogen receptor β, integrin α5, integrin β1, Rac1, RhoA, vascular endothelial growth factor α (VEGFα), kidney injury molecule 1 (KIM1), neutrophil gelatinase-associated lipocalin (NGAL) and glyceraldehyde-3-phosphate dehydrogenase (GAPDH) (Applied Biosystems, Foster City, CA) was determined using assay-on-demand primers/probe sets (TaqMan Gene Expression, Thermo Fisher Scientific, Bleiswijk, The Netherlands; Table 1). Gene expression was normalized to the expression of the housekeeping gene GAPDH, yielding the ΔCT value.

## Histology

Four-micrometer-thick paraffin sections of lung and kidney tissue were stained with hematoxylin and eosin. Pulmonary sections were scored in a semi-quantitative fashion on a scale of 0–3 for the following parameters: perivascular edema, alveolar edema and interstitial inflammation. Kidney tissue was assessed for interstitial edema and signs of tubular injury on a scale of 0–3. A score of 0 represented normal tissue, less than 25% presence of above described parameters; 1 represented mild, 25 to 50%; 2 represented moderate, 50 to 75%; 3 represented severe, more than 75% presence.

## Statistical analysis

The normality of data distribution was determined using the Shapiro-Wilk test. As data appeared non-normally distributed, all data are expressed as median with interquartile range and analyzed using GraphPad Prism 8.0 (GraphPad Software, La Jolla, CA, USA). Primary outcome was renal edema formation. At least an increase in wet/dry weight ratio from $3.6 \pm 0.4$ to 4.1 was expected based on pilot experiments. With a two-sided significance level (α) of 0.05 and β of 0.9, group sizes of 9 mice were calculated to reach statistical significant differences. For every experimental group, all 9 mice were included for analysis. Sex differences in male and female Tie2$^{+/+}$ and Tie2$^{+/-}$ mice were evaluated by Kruskal-Wallis testing with Dunn's post hoc analyses. P values less than 0.05 were considered as statistically significant.

The experimental data that support the findings of this study are available in DANS EASY with the identifier https://doi.org/10.17026/dans-zea-fs7p [28].

## Results

### Sex differences in wild type mice with normal Tie2 expression

We first investigated whether sex-related differences exist in mice with normal Tie2 expression by comparing male and female Tie2$^{+/+}$ mice. In general, female mice had a significantly lower body weight (23.0 [21.5–24.2] vs. 27.9 [26.7–36.3] gram, p = 0.0031) compared to male mice.

To provide insight in basal expression levels of the endothelial angiopoietin/Tie2 system, we determined whether circulating and renal and pulmonary gene expression levels of key molecules involved in endothelial angiopoietin/Tie2 signaling were differently expressed between males and females. In wild-type mice, circulating levels of angiopoietin-1 (p = 0.06) and angiopoietin-2 were higher in females compared to males (Fig 2A and 2B), whereas angiopoietin-2/angiopoietin-1 ratio or soluble Tie2 levels were comparable between males and females (Fig 2C and 2D).

Renal gene expression of angiopoietin-1 and angiopoietin-2 was higher in females compared to males (Fig 3A and 3B). Renal gene expression of Tie2 and Tie1 was also higher in females compared to males, however, both did not reach statistical significance (both p = 0.09, Fig 3C and 3D), whereas renal VE-PTP gene expression did not differ between female and male mice (Fig 3E). No differences were found between male and female mice with regard to pulmonary gene expression of angiopoietin-1, angiopoietin-2, Tie2, Tie1 or VE-PTP (Fig 4A–4E).

Next, we have investigated whether basal differences exist between male and female mice in commonly used markers of organ injury, such as renal and pulmonary edema formation and renal ischemia (NGAL and KIM-1). Renal and pulmonary wet/dry weight ratio, as measure of organ edema, did not differ between female and male mice (Fig 5A and 5B). Basal circulating levels of NGAL were lower in females compared to males (Fig 5E). Renal gene expression of NGAL was higher and renal gene expression of KIM1 was lower in females compared to males, however, both did not reach statistical significance (p = 0.06 and p = 0.09, respectively; Fig 5F and 5G). Histopathological analysis revealed no renal or pulmonary damage for both sexes (Fig 5C and 5D).

As a first step in finding an explanation for the sex-differences in the endothelial angiopoietin/Tie2 system, we determined renal and pulmonary estrogen gene expression levels. Renal and pulmonary gene expression of estrogen receptor α was comparable between males and females (Figs 6A and 7A). Gene expression of estrogen receptor β was higher in female than in male kidneys (Fig 5B), whereas no differences were found between sexes in lungs (Fig 7B).

Additionally, as we found differences in basal expression levels in the angiopoietin/Tie2 system, we were interested whether other (downstream) endothelial barrier regulators differed between sexes due to the complex interaction with the endothelial angiopoietin/Tie2 system. Renal gene expression of vascular endothelial growth factor (VEGF) α was higher in females compared to males (Fig 6E). Renal gene expression of integrin α5, integrin β1, RhoA and Rac1 did not differ between females and males (Fig 6C, 6D, 6F and 6G). In pulmonary tissue, no differences in gene expression were found between males and females for integrin α5, integrin β1, VEGFα, RhoA and Rac1 (Fig 7C–7G).

### Sex differences in heterozygous Tie2$^{+/-}$ mice with partial deletion of Tie2

Following the confirmation of sex-related differences in basal expression of molecules involved in endothelial angiopoietin/Tie2 signaling, we subsequently investigated the effect of sex in

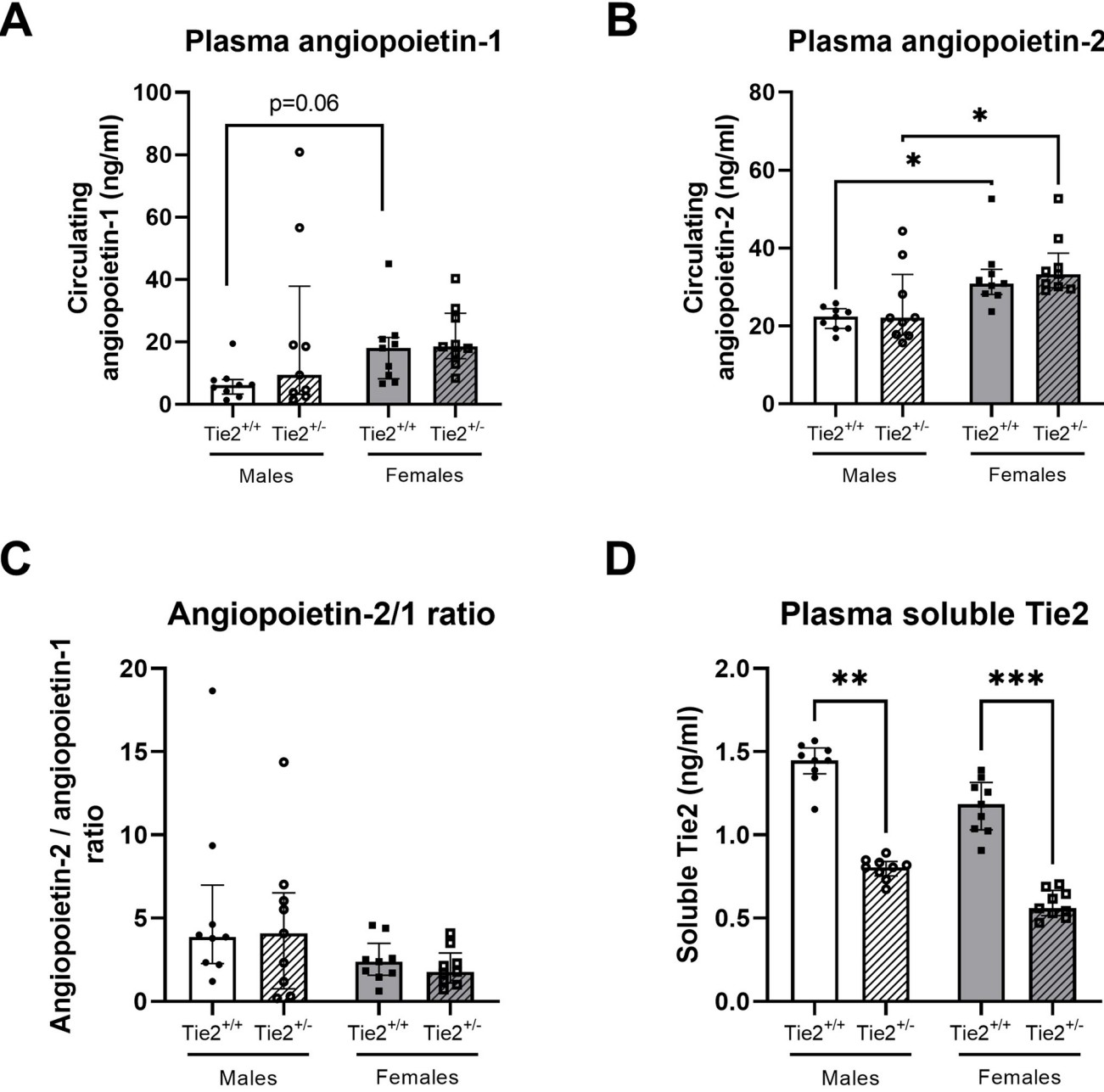

**Fig 2. Circulating markers of the endothelial angiopoietin/Tie2 system.** Circulating levels of angiopoietin-1 (**A**), angiopoietin-2 (**B**), angiopoietin-2/1 ratio (**C**) and soluble Tie2 (**D**) in plasma of Tie2$^{+/+}$ male (white bars), Tie2$^{+/+}$ female (grey bars), Tie2$^{+/-}$ male (striped white bars) and Tie2$^{+/-}$ female (striped grey bars) mice. Each dot represents an individual mouse. Data are presented as median with interquartile range. * $p<0.05$, ** $p<0.01$, *** $p<0.001$.

heterozygous mice with lower basal endothelial Tie2 expression (Tie2$^{+/-}$), thereby mimicking the suppressive effect of THS on the endothelial Tie2 receptor. First, we confirmed the heterozygous Tie2 knockout by determining renal and pulmonary gene expression levels of Tie2. Renal Tie2 gene expression was reduced by 48% in heterozygous male and 60% in heterozygous female mice (Fig 3C). Pulmonary Tie2 gene expression was reduced by 51% in heterozygous male and 41% in heterozygous female mice (Fig 4C).

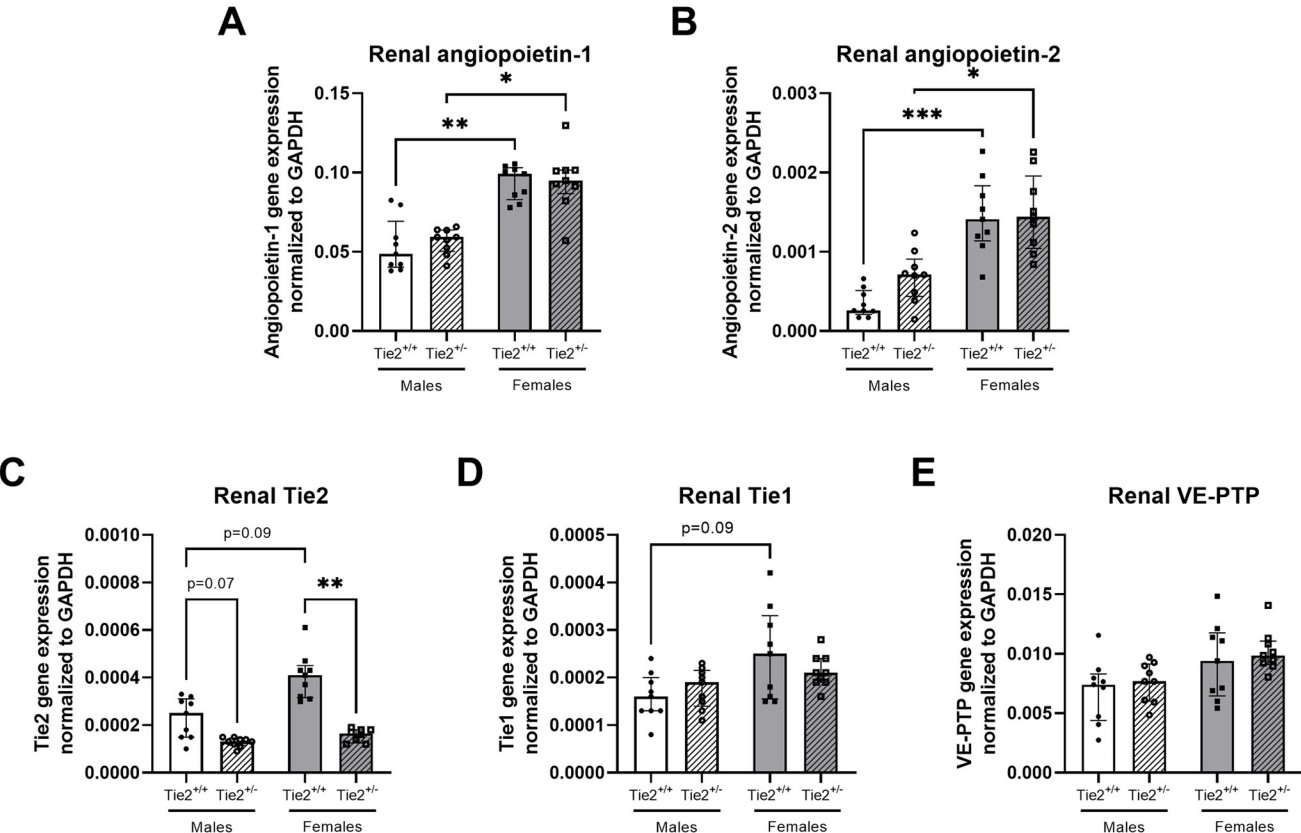

**Fig 3. Renal gene expression of the endothelial angiopoietin/Tie2 system.** Renal gene expression of angiopoietin-1 (**A**), angiopoietin-2 (**B**), Tie2 (**C**), Tie1 (**D**) and vascular endothelial protein tyrosine phosphatase (VE-PTP; **E**) in Tie2$^{+/+}$ male (white bars), Tie2$^{+/+}$ female (grey bars), Tie2$^{+/-}$ male (striped white bars) and Tie2$^{+/-}$ female (striped grey bars) mice. Each dot represents an individual mouse. Data are presented as median with interquartile range. * p<0.05, ** p<0.01, *** p<0.001.

Next, we identified whether Tie2 heterozygosity resulted in differences in body weight between males and females. In Tie2$^{+/-}$ mice, body weight was similar to Tie2$^{+/+}$ controls for both males (31.1 [30.2–33.5] vs. 27.9 [26.7–36.3] gram, p>0.99) and females (24.6 [21.9–28.3] vs. 23.0 [21.5–24.2] gram, p = 0.79). Similar to Tie2$^{+/+}$ controls, Tie2$^{+/-}$ females had a lower body weight compared to Tie2$^{+/-}$ males (24.6 [21.9–28.3] vs. 31.1 [30.2–33.5] gram, p = 0.05).

We also determined expression of molecules involved in endothelial angiopoietin/Tie2 signaling in heterozygous knockout of Tie2 and whether differences exist between males and females. Circulating levels of angiopoietin-1 did not differ between Tie2$^{+/+}$ and Tie2$^{+/-}$ mice in both males and females (Fig 2A). Likewise, circulating plasma levels of angiopoietin-2 were not affected by heterozygosity, nevertheless they were higher in Tie2$^{+/-}$ females compared to Tie2$^{+/-}$ males (Fig 2B). This resulted in similar angiopoietin-2/angiopoietin-1 ratios in both sexes, which were not affected by partial deletion of Tie2 (Fig 2C). Soluble Tie2 levels were decreased in both female and male Tie2$^{+/-}$ mice compared to Tie2$^{+/+}$ female and male mice (Fig 2D).

Renal gene expression of angiopoietin-1, angiopoietin-2, Tie1 and VE-PTP did not differ between Tie2$^{+/+}$ and Tie2$^{+/-}$ in both female and male mice (Fig 3A, 3B, 3D and 3E), whereas renal angiopoietin-1 and angiopoietin-2 gene expression was higher in Tie2$^{+/-}$ females compared to Tie2$^{+/-}$ males (Fig 3A and 3B). No differences in pulmonary gene expression were

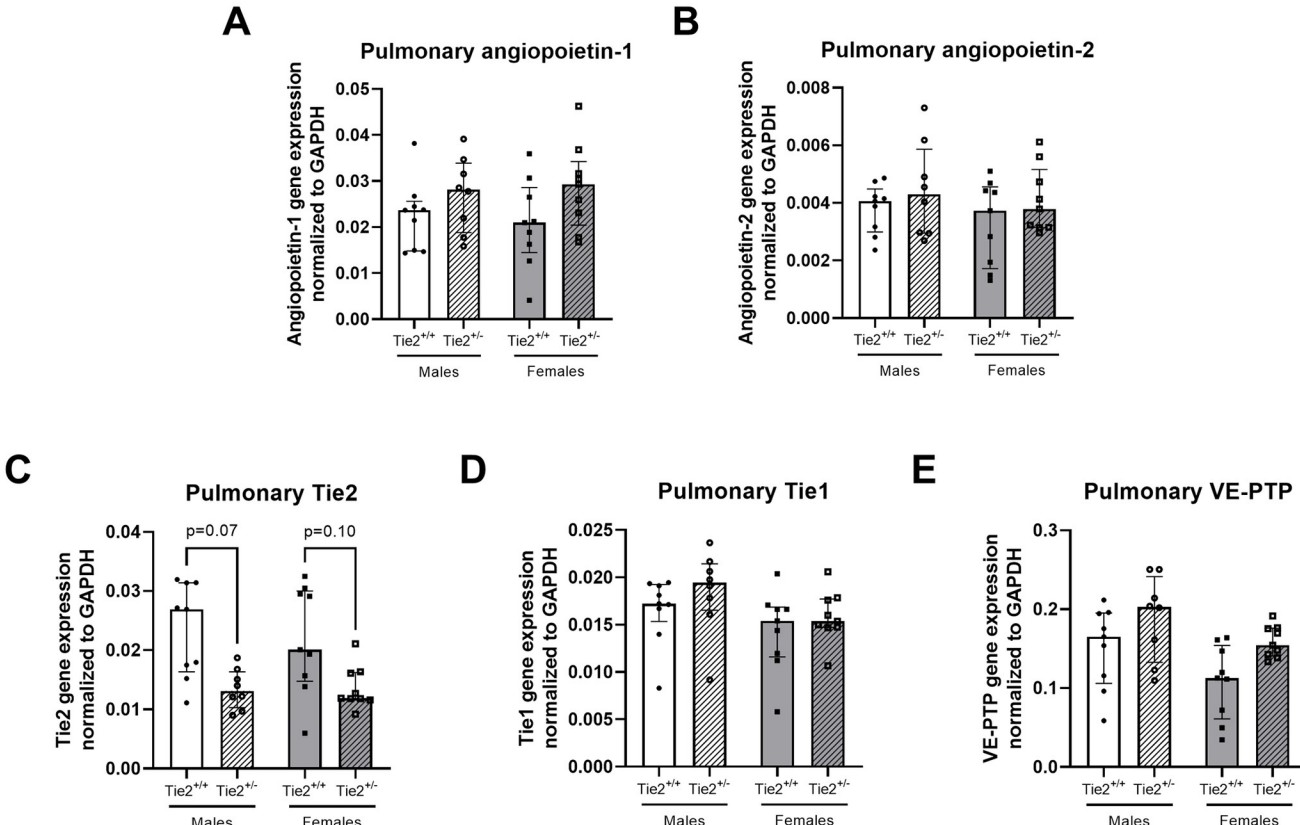

**Fig 4. Pulmonary gene expression of the endothelial angiopoietin/Tie2 system.** Pulmonary gene expression of angiopoietin-1 (**A**), angiopoietin-2 (**B**), Tie2 (**C**), Tie1 (**D**) and vascular endothelial protein tyrosine phosphatase (VE-PTP; **E**) in Tie2$^{+/+}$ male (white bars), Tie2$^{+/+}$ female (grey bars), Tie2$^{+/-}$ male (striped white bars) and Tie2$^{+/-}$ female (striped grey bars) mice. Each dot represents an individual mouse. Data are presented as median with interquartile range.

found for angiopoietin-1, angiopoietin-2, Tie1 or VE-PTP between Tie2$^{+/+}$ and Tie2$^{+/-}$ in both female and male mice (Fig 4A, 4B, 4D and 4E).

As THS is associated with reduced Tie2 expression and edema formation in kidneys and lungs, we subsequently investigated whether partial deletion of Tie2 resulted in renal or pulmonary edema and whether sex-related differences exist in renal or pulmonary edema and expression of renal ischemia markers in mice with lowered Tie2 expression. Tie2$^{+/-}$ males had higher renal wet/dry weight ratios compared to Tie2$^{+/+}$ males (Fig 5A). Interestingly, this higher renal wet/dry weight ratio was not observed in female Tie2$^{+/-}$ mice compared to Tie2$^{+/+}$ females (Fig 5A). In lungs, no differences in wet/dry weight ratio were found between all groups (Fig 5B). Histopathological analysis revealed no renal or pulmonary damage for all groups (Fig 5C and 5D). Circulating levels of NGAL did not differ between Tie2$^{+/-}$ and Tie2$^{+/+}$ mice in both males and females (Fig 5E). However, female Tie2$^{+/-}$ mice had significantly lower circulating levels of NGAL compared to Tie2$^{+/-}$ males (Fig 5E). Renal gene expression of NGAL and KIM1 did not differ between Tie2$^{+/-}$ and Tie2$^{+/+}$ mice of both sexes (Fig 5F and 5G), whereas female Tie2$^{+/-}$ mice had significantly lower gene expression levels of KIM1 compared to Tie2$^{+/-}$ male mice (Fig 5G).

As a first step in finding an explanation for the suggested protective effect of female sex in renal, but not pulmonary edema formation in heterozygous Tie2 knockout mice, we

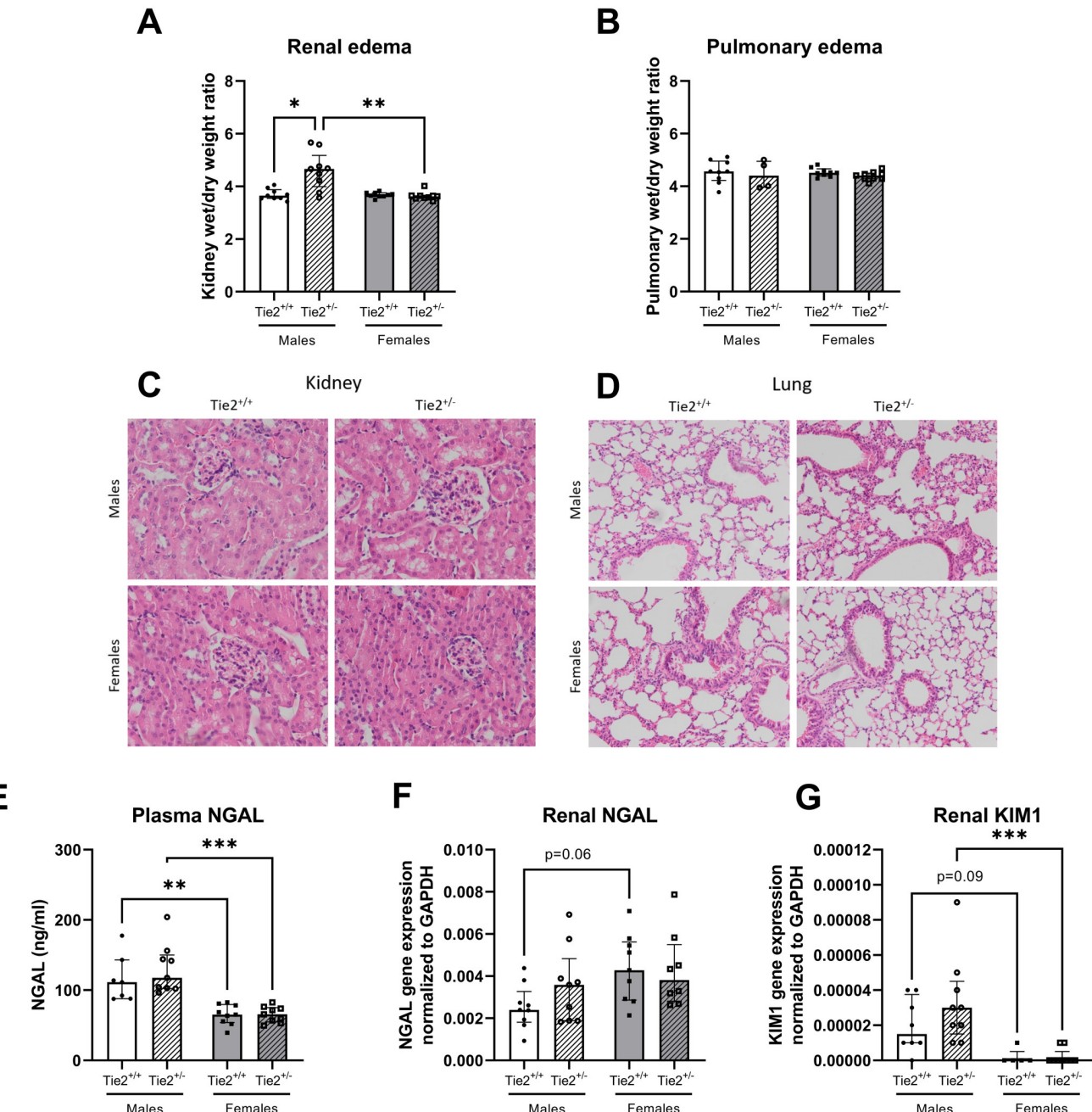

**Fig 5. Sex differences and effect of Tie2 heterozygosity in renal and pulmonary function.** Renal wet/dry weight ratio (**A**), pulmonary wet/dry weight ratio (**B**), H&E stainings of one typical example of each group in kidney (**C**) and lung (**D**), circulating plasma levels of neutrophil gelatinase-associated lipocalin (NGAL; **E**) and renal gene expression of NGAL (**F**) and kidney injury molecule 1 (KIM1; **G**) in Tie2+/+ male (white bars), Tie2+/+ female (grey bars), Tie2+/- male (striped white bars) and Tie2+/- female (striped grey bars) mice. Each dot represents an individual mouse. Data are presented as median with interquartile range. * p<0.05, ** p<0.01, *** p<0.001.

determined whether expression levels of estrogen receptors and receptors that closely interact with the endothelial angiopoietin/Tie2 system differ between male and female Tie2+/- mice. Renal gene expression levels of estrogen receptor β did not differ between Tie2+/- and Tie2+/+ mice, but were higher in female Tie2+/- mice compared to Tie2+/- male

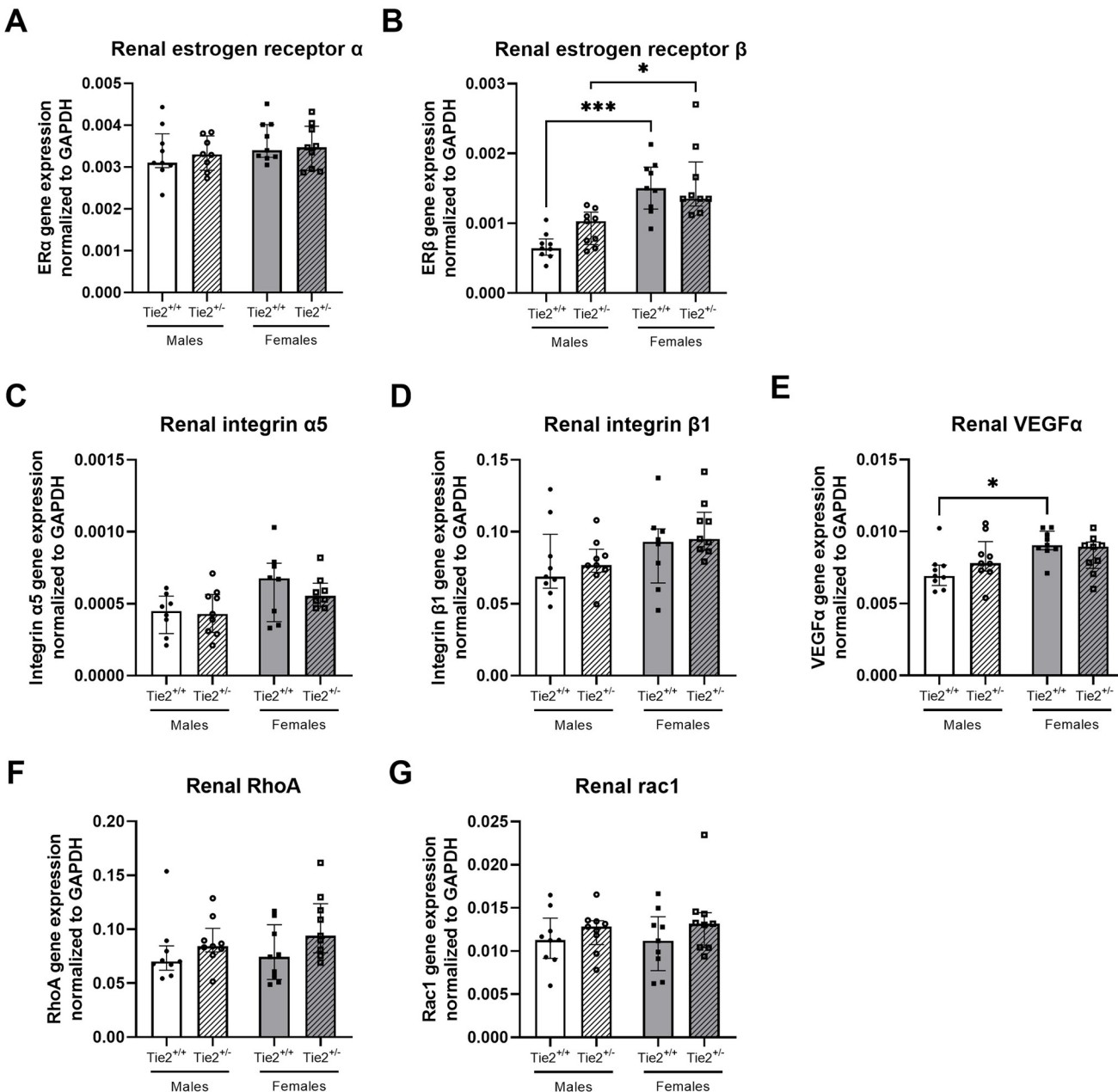

**Fig 6. Renal gene expression of other endothelial barrier regulators.** Renal gene expression of estrogen receptor α (**A**), estrogen receptor β (**B**), integrin α5 (**C**), integrin β1 (**D**), vascular endothelial growth factor α (VEGFα; **E**), RhoA (**F**) and Rac1 (**G**) in Tie2$^{+/+}$ male (white bars), Tie2$^{+/+}$ female (grey bars), Tie2$^{+/-}$ male (striped white bars) and Tie2$^{+/-}$ female (striped grey bars) mice. Each dot represents an individual mouse. Data are presented as median with interquartile range. * p<0.05, *** p<0.001.

mice (Fig 6B), whereas expression of estrogen receptor α was comparable (Fig 6A). No differences were found in pulmonary gene expression of estrogen receptor α or β (Fig 7A and 7B). Renal gene expression of integrin α5, integrin β1, VEGFα, RhoA and Rac1 was not affected by sex and did not differ between Tie2$^{+/-}$ and Tie2$^{+/+}$ mice (Fig 6C–6G). In lungs, RhoA gene expression was higher in female Tie2$^{+/-}$ mice compared to female Tie2$^{+/+}$ mice (Fig 7F), whereas gene expression of integrin α5, integrin β1, VEGFα, and

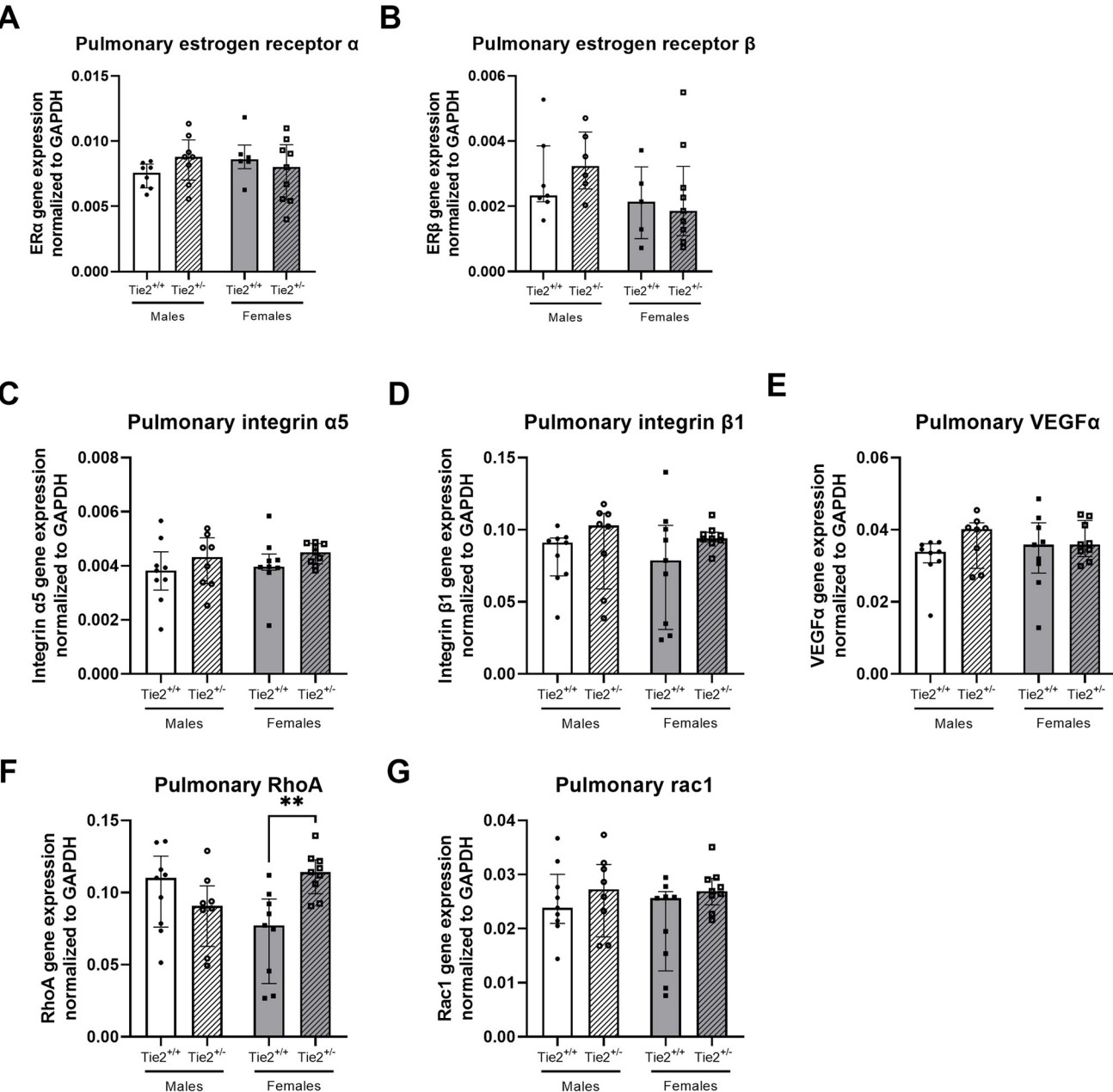

**Fig 7. Pulmonary gene expression of other endothelial barrier regulators.** Pulmonary gene expression of estrogen receptor α (**A**), estrogen receptor β (**B**), integrin α5 (**C**), integrin β1 (**D**), vascular endothelial growth factor α (VEGFα; **E**), RhoA (**F**) and Rac1 (**G**) in Tie2$^{+/+}$ male (white bars), Tie2$^{+/+}$ female (grey bars), Tie2$^{+/-}$ male (striped white bars) and Tie2$^{+/-}$ female (striped grey bars) mice. Each dot represents an individual mouse. Data are presented as median with interquartile range. ** p<0.01.

Rac1 was not affected by sex and did not differ between Tie2$^{+/-}$ and Tie2$^{+/+}$ mice (Fig 7C–7E and 7G).

## Discussion

Female sex is suggested to be a potential advantage in outcome following THS [1–5]. Endothelial hyperpermeability contributes to THS-induced acute kidney and lung injury. Endothelial

permeability is, amongst others, regulated by the endothelial angiopoietin/Tie2 system. During THS, the endothelial angiopoietin/Tie2 system is disbalanced and pharmacological activation of Tie2 reduces THS-induced vascular leakage in males. However, sexual dimorphism of the endothelium has not been taken into account. In the current study, we investigated whether sex-related differences exist in basal expression of the endothelial angiopoietin/Tie2 system. Additionally, we investigated whether reduced Tie2 expression, as seen during THS, resulted in a sex-dependent change in expression levels of the endothelial angiopoietin/Tie2 system and whether it results in renal or pulmonary injury. To our knowledge, we are the first to show the existence of sexual dimorphism in the basal expression levels of key molecules involved in endothelial angiopoietin/Tie2 signaling in healthy mature mice as reflected by higher expression of angiopoietin-1 and angiopoietin-2 in plasma and kidney, but not lungs, in females compared to males. Genetically reduced Tie2 expression did not affect proteins and genes involved in the endothelial angiopoietin/Tie2 system. Interestingly, heterozygous male mice with partial deletion of Tie2 showed renal edema, while this was not observed in females. These differences could not be explained by sex-related differences in the endothelial angiopoietin/Tie2 system. Interestingly, pulmonary edema was comparable between all groups and no difference in angiopoietin/Tie2 expression was found in lungs, which suggests also organ-specific regulations of endothelial angiopoietin/Tie2 signaling. Future studies are warranted to elaborate on the relation between sex differences and angiopoietin/Tie2 signaling in the development of organ edema.

The endothelial angiopoietin/Tie2 system is involved in the regulation of endothelial permeability, and it is not known whether sex affects expression of components of this system. In the present study, we showed that sex differences in basal expression levels of the endothelial angiopoietin/Tie2 system exist. Circulating levels of angiopoietin-2 were higher and soluble levels of Tie2 were lower in females compared to males. This is in agreement with the findings of a community-based cohort study of almost 4000 participants in which females also showed higher circulating angiopoietin-2 and lower soluble Tie2 levels compared to males [29]. In our study, both circulating angiopoietin-1 and angiopoietin-2 levels were higher in females compared to males and therefore the angiopoietin-2/angiopoietin-1 ratio was comparable between males and females, suggesting that Tie2 is activated to a similar extend by the angiopoietins in males and females. However, lower soluble Tie2 levels in females suggest less shedding of the Tie2 receptor, while increased shedding has previously been linked to inflammation and complement activation in septic patients [30]. As far as we know, we are the first to show sexual dimorphism of the endothelial angiopoietin/Tie2 system in healthy conditions. Sexual dimorphism in the endothelial angiopoietin/Tie2 system and corresponding explanations are limitedly reported. One of the basal key differences between females and males are sex hormones, which most often emerge around the time of sexual maturation. Interestingly, a potential advantage of female gender in THS seems specifically during their reproductive phase [2, 5, 6], suggesting a role for estrogen. Due to limited plasma volumes, we were not able to measure circulating estrogen in the current study, but based on the age of mice we expect that estrogen levels were higher in female compared to male mice [27, 31]. Estrogens may regulate the endothelial angiopoietin/Tie2 system, however, contrasting results exist. As determined in the endometrium of ovariectomized animals, systemic administration of estrogen increases gene expression of angiopoietin-1, but not angiopoietin-2 or Tie2 [32], whereas longer exposure to estrogen reduces angiopoietin-1 and Tie2, but increased angiopoietin-2 expression [33]. In renal tissue of ovariectomized rats, Ye *et al.* showed that estrogen reduced angiopoietin-1 and increased angiopoioetin-2 gene expression, but did not affect Tie2 gene expression [34]. These apparent conflicting results might be explained by differences in exposure time to estrogen, organ systems and the fact that only ovariectomized animals were studied. In the current study

we report higher renal estrogen receptor β expression in females, which is previously shown to increase angiopoietin-2 expression [35]. Interestingly, we found no differences between males and females in endothelial angiopoietin/Tie2 signaling in lungs, suggesting that sex mainly affects renal angiopoietin/Tie2 signaling. Taken together, our results suggest that sexual dimorphism of the endothelial angiopoietin/Tie2 system exists in kidneys of healthy mice. Further research on the role of estrogen in the regulation of the endothelial angiopoietin/Tie2 system is of interest.

THS induces systemic inflammation, leading to a permeable endothelium, tissue edema and organ injury [9, 10]. We and others have previously shown reduced endothelial Tie2 expression, endothelial hyperpermeability and organ injury in animals following HS [10, 12, 21] and sepsis [21, 30] and in postmortem renal tissue of septic patients [36]. To study the effect of reduced Tie2 expression without additional effects such as systemic inflammation, we have included heterozygous Tie2 knockout mice. As expected, we found renal edema in male mice with partial deletion of Tie2. This is in line with previous studies where suppression of Tie2 induced endothelial hyperpermeability in an *in vitro* setting [37]. Against our expectations, we found no pulmonary edema in male mice with partial deletion of Tie2. Although previous studies described the occurrence of pulmonary edema [37] and decreased pulmonary microcirculatory perfusion *in vivo* [38] in response to reduced Tie2 expression, this was all initiated by either sepsis [36] or pulmonary arterial hypertension [38], suggesting pulmonary edema only occurs in response to both reduced Tie2 expression and an additional inflammatory hit. Surprisingly, we have found that female mice with partial deletion of Tie2 did not have renal or pulmonary edema formation. This absence of renal edema in heterozygous Tie2 knockout mice cannot be explained by differences in the extent of Tie2 knockout as female mice tended to have an even bigger reduction in Tie2 gene expression. A possible explanation is that in healthy humans, mean arterial pressure, oncotic pressure and capillary pressure are lower in premenopausal females compared to males, which will offset a higher fluid movement out of the capillaries in males [39, 40]. Additionally, differences exist between male and female endothelial cells in the *in vitro* setting [41, 42]. Male endothelial cells express less vascular endothelial cadherin (VE-cadherin), a protein regulating cell-cell interactions, and form a less strong endothelial barrier layer compared to female endothelial cells [42]. In addition, female sex hormones have a protective role in THS-induced pulmonary leakage [43] and lung injury [44]. However, above described differences have not been studied in different organs. Taken together, female mice appear to have a protective mechanism against renal edema formation following partial deletion of endothelial barrier regulator Tie2. Sex-related endothelial dysfunction receives insufficient attention and further research is warranted to determine sexual dimorphism of the endothelium.

Female gender seems also protective in the development of renal failure [2, 5, 7] and the progression to end-stage renal failure [45]. Structural differences exist between males and females, as males for example have a larger kidney with larger glomeruli volume compared to females [46]. Moreover, sex hormones regulate several cellular processes that affect renal function, such as the release of cytokines and synthesis of proteins. For example, estradiol may protect against the progression of renal disease by inducing nitric oxide synthesis in glomerular endothelial cells [47]. In the current study, we have provided evidence for differences in basal expression of markers commonly used for renal ischemia as females presented lower circulating NGAL and expressed lower KIM1 compared to males. These results confirm the findings of another mouse study investigating the effect of sex on cisplatin-induced acute kidney injury, where females presented lower renal KIM-1 and circulating NGAL expression [48]. In summary, we have shown sex-related differences in the basal expression levels of commonly used renal ischemia markers, which should be further explored and confirmed in the clinical situation.

## Limitations

The current study was designed to provide a first overview on sex-related differences in the expression levels of the endothelial angiopoietin/Tie2 system. Our findings significantly increase basic knowledge, however, it would be of interest to further investigate these parameters in response to THS, with corresponding effects on activation of the Tie2 receptor, to increase translatability to the clinical setting. Additionally, due to absence of specific antibodies to determine phosphorylation of Tie2 in mice and the limited amount of tissue available, the current study did not investigate the functional aspects of the endothelial angiopoietin/Tie2 system. The current study was carried out on breeding surplus, thereby contributing to the reduction and refinement of animal studies. However, a study with breeding surplus is limited to only using non-invasive techniques. As the current study showed interesting sex-related differences, future studies could investigate these differences more in depth by for example increasing or reducing estrogen levels and thereby study the relation between estrogen and Tie2 more closely.

## Conclusions

The present study showed sexual dimorphism of the endothelial angiopoietin/Tie2 system, which is involved in the regulation of endothelial permeability. Additionally, the current study revealed a sex-related susceptibility to reduced basal Tie2 expression. Female sex protects against renal edema in mice with partial deletion of Tie2. Future studies should reveal whether comparable sex-related differences exist in humans, and whether these differences contribute to sex-related susceptibility to renal injury. As a relation between estrogen and the endothelial angiopoietin/Tie2 system is suggested, it would be of interest to further explore this relation. Upon confirmation of the findings of the current study in THS patients, these differences may be the basis of the development of sex-specific treatment strategies to improve outcome following traumatic hemorrhagic shock.

## Supporting information

**S1 Fig. Typical example of genotyping results.**
(TIF)

## Author Contributions

**Conceptualization:** Anoek L. I. van Leeuwen, Nicole A. M. Dekker, Charissa E. van den Brom.

**Data curation:** Anoek L. I. van Leeuwen, Roselique Ibelings, Charissa E. van den Brom.

**Formal analysis:** Anoek L. I. van Leeuwen, Marjolein R. A. van der Steen, Joris J. T. H. Roelofs.

**Funding acquisition:** Nicole A. M. Dekker, Charissa E. van den Brom.

**Investigation:** Anoek L. I. van Leeuwen, Nicole A. M. Dekker.

**Methodology:** Anoek L. I. van Leeuwen, Nicole A. M. Dekker, Marjolein R. A. van der Steen, Joris J. T. H. Roelofs.

**Project administration:** Roselique Ibelings.

**Supervision:** Charissa E. van den Brom.

**Validation:** Matijs van Meurs, Grietje Molema.

**Visualization:** Anoek L. I. van Leeuwen.

**Writing – original draft:** Anoek L. I. van Leeuwen, Elise Beijer, Charissa E. van den Brom.

**Writing – review & editing:** Anoek L. I. van Leeuwen, Elise Beijer, Matijs van Meurs, Grietje Molema, Charissa E. van den Brom.

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
