## [Decision Letter · Decision Letter 0]

14 Jun 2023

PONE-D-23-11857Female sex protects against renal edema in mice with partial deletion of the endothelial barrier regulator Tie2PLOS ONE

Dear Dr. van den Brom,

Thank you for submitting your manuscript to PLOS ONE. After careful consideration, we feel that it has merit but does not fully meet PLOS ONE’s publication criteria as it currently stands. Therefore, we invite you to submit a revised version of the manuscript that addresses the points raised during the review process.

We look forward to receiving your revised manuscript.

Kind regards,

Keiko Hosohata, Ph.D.

Academic Editor

PLOS ONE

Journal Requirements:

Reviewers' comments:

Reviewer's Responses to Questions

**Comments to the Author**

1. Is the manuscript technically sound, and do the data support the conclusions?

Reviewer #1: Partly

Reviewer #2: Yes

Reviewer #3: Partly

Reviewer #4: Yes

Reviewer #5: No

Reviewer #6: Yes

2. Has the statistical analysis been performed appropriately and rigorously? 

Reviewer #1: No

Reviewer #2: Yes

Reviewer #3: I Don't Know

Reviewer #4: Yes

Reviewer #5: Yes

Reviewer #6: Yes

3. Have the authors made all data underlying the findings in their manuscript fully available?

Reviewer #1: No

Reviewer #2: Yes

Reviewer #3: No

Reviewer #4: Yes

Reviewer #5: No

Reviewer #6: Yes

4. Is the manuscript presented in an intelligible fashion and written in standard English?

Reviewer #1: No

Reviewer #2: Yes

Reviewer #3: Yes

Reviewer #4: Yes

Reviewer #5: No

Reviewer #6: Yes

5. Review Comments to the Author

Reviewer #1: 1. The little should be modified to include male and female mice.

2. There is need for critical grammatical review.

3. Wet/dry weight ratio is not the ideal test for renal and pulmonary injury, but an indicator for renal and pulmonary edeme. Medications, injections, etc can also cause edema.

4. Abstract should be revised to give a comprehensive summary of background, method, result and conclusion.

5. Conclusion shows that female mice could protect against renal edema, though the mechanism could not be explained. The preresent finding is a clear confirmation of the established findings.

6. Angiopoeitin-1 is a growth factor produced by kidney and liver, whereas angiopoeitin-2is produced and stored in Weibel-Palade bodies in endothelial cells. It binds to integrin beta-2 by interacting with platelete-derived growth factor –BB in monocytes. Hence determination of integrin and PLDGF-BB is necessary. Angiopoetin-1 signals through Tie 2, whereas angiopoetin-2 is antagonistic, hence it is a regular disruptive agent. The study was on both kidney and lung of the rats, therefore the tittle does not reflect the content.

7. Introduction should be revised to include the existing discoveries, shortcomings and the way forward. The aim of the study should be redefined. The emphasis is on the kidney and the study on both lung and kidney. Liver should have been included.

8. The methods are technical but grossly inadequate. Source, age, weight and strain of the mice are not mentioned. Period of acclimatization; cyclical diurnal changes, feeds, period of experimentation, vital parameters and generation of the mice (1st filial or 2nd filial) are not mentioned. Euthanasia using isoflurane is controversial. It is not the best drug to use. It causes behavioral changes such as induction of attention deficit, increased anxiety etc. Subheadings under Methods should be rearranged as follows: Experimental set up, Animals and Genotyping, Renal and pulmonary edema formation, Plasma analysis, Renal analysis and study approval. The use of isoflurane and decapitation is a great cruelty to Animal kingdom especially the species of mice. It is quite disheartening. Inhaled isoflurane cause pulmonary edema by atenuating histologic lung injury. I am quite wary of the use of isoflurane/decapitation. Note that isoflurane (5%) can cause death in 60 sec (too long) for humane killing. Respiratory arrest takes longer time.

9. Statistical methods should be revised. All the data presented in barcharts should be revised. Some data generated are quantitative while others are qualitative. The conclusion derived from the study based on the result is quite unfounded. The presented results are not adequately comprehensive. Therefore, I am of opinion that some results should be changed to tabular form, since males and females were used for the study. Quantitative data could be analyzed using student test unpaired. Specific relevant changes in the gene of each mice should be computed and the differences in the genes could be ranked. The Results section requires critical grammatical review. Functional parameters of kidney should have been measured.

10. Discussion should be revised. Findings should be discussed with reference to the past findings.

11. Conclusion is grossly inadequate and biased. Hence kindly revised the conclusion.

Reviewer #2: Dear

1. Introduction. It is informative with compatible references

2. Methods: other than the sample size are small, it is clear and concise. Small sample size may bias the results.

3. Procedures of methods: clear and concise

4. Statistical analysis: I think with a small number of animals, it is difficult to get statistically significant.

5. The results were presented as figures, which appeared a little bit confused. The other option is to tabulate some of your findings , which be of high values to the readers as well as you impressed them.

6. Please, revised your conclusion to be more informative.

Reviewer #3: Title: Female sex protects against renal edema in mice with partial deletion of the endothelial barrier regulator Tie2

Authors: N/A

Review

Comments to Author: The role of biological sex as a risk factor for adverse outcomes following major trauma has been widely recognized. Females may confer a protective effect against organ failure, sepsis, and mortality in patients experiencing traumatic hemorrhage. The endothelial angiopoietin/Tie2 system is well recognized for controlling the permeability of endothelial cells. In this study, in heterozygous Tie2 knock-out mice, it appears that the female sex provides protection against renal edema while not exerting the same protective effect against pulmonary edema. This study suggests that there is a sexual dimorphism that provides protection against renal edema in female mice. The study indicated that Tie2 has exhibited potential in mitigating organ edema induced by hemorrhagic shock, specifically in males. However, it is imperative to acknowledge the potential existence of sexual dimorphism in the endothelium, an aspect that has not been investigated in this particular context before.

Major Comments:

• The authors of the paper have done an impressive job in carrying out their research and presenting their findings. They demonstrated a good level of skill and attention to detail in executing the study and analyzing the data. Additionally, they extensively reviewed the existing literature related to their research topic, providing a comprehensive understanding of the subject matter.

• However, it is important to note that the experimental design employed in the study could have been more carefully chosen. This means that there may be some limitations in how the study was designed. These flaws could potentially impact the reliability of the results obtained.

• The authors investigated "whether expression of components of the endothelial angiopoietin/Tie2 system in kidneys and lungs differs between healthy female and male mice. The effect of sex on renal and pulmonary injury in heterozygous mice with genetically reduced endothelial Tie2 expression, thereby mimicking the suppressive effect of THS on expression of the endothelial Tie2 receptor."

• Although the tie-2 deletion has been universal, the authors failed to explain the organ-specific effect of this particular phenotype.

• The discrepancy between gene and plasma expression has not been discussed.

• The abstract stated "estrogens and other endothelial barrier regulators," although only estrogen receptors have been analyzed. While estrogen has been briefly discussed with regards to its role in vasoprotective effects, the postulated selective role of TIE-2 has not been optimally elucidated in this paper.

• The sample size is relatively small for assessing the claimed hypotheses.

• Ln 159: Histology has not been presented anywhere in the present study, and the removal of figures has not been justified.

Minor Comments:

o Reference [5] are repeated many times.

o A pharmacological blocker could have a superior outcome, although this has not been utilized in the study and previously utilized by your group

o Ln 81, typo error

o The wet/dry weight ratio approach did not show any significant difference in the present study. I wonder if there is a better mechanistic approach to show the investigated phenotype (lung vs. renal edema).

o Ln 188: "Circulating levels of angiopoietin-1 (p=0.06) and angiopoietin-2 were higher in females compared to males (Fig 1A-B)" is not entirely supported by the present findings and referenced figures

o Ln 199 analyzed soluble tie2, I wondered if the cellular lysate of the compared organs might show a better conclusion.

o Ln 202 *, **, and *** symbols have not been included in the study.

o Not all selected genes are relevant to the proposed mechanism.

o Protein expression methods other than ELISA might also provide additional insight (e.g. western blotting).

o The selected cohort of genes is not fully justified. The molecules included in the analysis were estrogen receptor α, estrogen receptor β, integrin α5, integrin β1, Rac1, RhoA, vascular endothelial growth factor α (VEGFα) (why not other isoforms), kidney injury molecule 1 (KIM1), and neutrophil gelatinase-associated lipocalin (NGAL).

o Estrogen roles have been suggested in many reports but were not sufficiently discussed

o TIE-2 has been proposed as a protective factor, although in vitro studies have been lacking.

Reviewer #4: Good work. The paper is well written and discussed found that Female sex seems to protect against renal, but not pulmonary edema in heterozygous Tie2 knock-out mice. This could not be explained by sex dimorphism in the endothelial angiopoietin/Tie2 system.

Reviewer #5: Leeuwen and colleagues have investigated the effect of gender on edema formation in mice with partial deletion of Tie2.

Major concerns:

1- The authors have to justify why they did not use mice with full Tie2. In addition, please include the genotyping bands of three random mice in each group (in the supp material).

2- The authors relied on the w/d ratio and certain markers to conclude the edema formation. This needs further investigations such as, the albumin contents of the kidney tissues and BAL fluid as well as IHC analysis of the kidney and lung tissue to evaluate the the thickness of interalveolar septa or the tubules in the kidney tissues.

3- Tie2 receptor known to be extensively regulated on post-translational level, thus, the tissue levels of Tie2 need to be analysed (by WB or ELISA).

Minor concerns:

1- please label in each figure the corresponding tissue (kidney, pulmonary...).

2- please show the IHC data (stated in the results as data not shown).

3- The authors can design a scheme to describe the proposed mechanism.

Reviewer #6: Thank you for nice work and research. I have couple of recommendations:

Line 72-82 Add a simplified and summarized figure about the angiopoietin/Tie2 signaling pathway

Line 305-307 rewrite the two sentences regarding the pulmonary and renal gene expression to avoid duplication

All Figs (1B, 1D, 2A-C, 4A, 4C, 4D, 5B, 5E, & 6F) need to be revised and fix typo of (1)

6. PLOS authors have the option to publish the peer review history of their article (what does this mean?). If published, this will include your full peer review and any attached files.

Reviewer #1: No

Reviewer #2: **Yes: **Marwan S.M. Al-Nimer

Reviewer #3: No

Reviewer #4: No

Reviewer #5: No

Reviewer #6: No

---

## [Author Response · Author response to Decision Letter 0]

15 Sep 2023

PONE-D-23-11857

Response to the reviewers

Reviewer #1: 

We would like to thank the reviewer for putting his/her time and care in providing valuable comments which have significantly improved the manuscript. We answered the comments below in a point-wise manner. Changes and additions in the manuscript based on the provided comments are shown via track changes.

1. The little should be modified to include male and female mice. 

We assume the reviewer means ‘title’ with the word ‘little’. We have added ‘lung edema’ as suggested under point 6 and ‘compared to male sex’ and adjusted the title as follows: “Female sex protects against renal edema, but not lung edema, in mice with partial deletion of the endothelial barrier regulator Tie2 compared to male sex”.

2. There is need for critical grammatical review. 

We have critically reviewed our manuscript and corrected the text if necessary.

3. Wet/dry weight ratio is not the ideal test for renal and pulmonary injury, but an indicator for renal and pulmonary edeme. Medications, injections, etc can also cause edema. 

We agree with the reviewer that an increase in wet/dry weight ratio is an indication of edema formation. It depends on the setting whether edema can be an indicator of organ injury. For example in patients with sepsis, edema is a known complication and indicates organ injury. However, we agree with the reviewer that there are also settings where edema formation is not a marker for organ injury, indeed when edema occurs in response to for example medications. Therefore in the current study we combined measurements of organ edema with markers known to indicate organ injury, such as NGAL and KIM-1, but also IHC analysis. However, as we agree with the reviewer that edema can be a marker for organ injury, but is not per definition a marker for organ injury, we carefully revised our manuscript and changed terminology to ‘edema formation’ instead of ‘organ injury’ when only results of wet/dry weight ratios were discussed.

4. Abstract should be revised to give a comprehensive summary of background, method, result and conclusion. 

We critically read our abstract and revised it where necessary.

5. Conclusion shows that female mice could protect against renal edema, though the mechanism could not be explained. The present finding is a clear confirmation of the established findings.

We thank the reviewer for this positive comment.

6. Angiopoeitin-1 is a growth factor produced by kidney and liver, whereas angiopoeitin-2 is produced and stored in Weibel-Palade bodies in endothelial cells. It binds to integrin beta-2 by interacting with platelete-derived growth factor –BB in monocytes. Hence determination of integrin and PLDGF-BB is necessary. Angiopoetin-1 signals through Tie 2, whereas angiopoetin-2 is antagonistic, hence it is a regular disruptive agent. 

We have to kindly disagree with the reviewer. Angiopoietin-1 is primary expressed by pericytes and as far as current knowledge on this protein not expressed by only the kidney and liver. Additionally, although the by the reviewer described mechanism of angiopoietin/Tie2 signaling is correct in some cases, angiopoietin-2 may also function as agonist, depending on the context (PMID: 27038015; PMID: 19223473). Lastly, to our knowledge, primary endothelial Tie2 signaling goes via integrin α5/β1, and not integrin β2. Therefore, we included analysis of integrin a5 and integrin b1 in our manuscript (figure 6 and 7). The interaction with integrin beta-2 that is described by the reviewer is present in monocytes, whereas the current study focused on endothelial angiopoietin/Tie2 signaling, therefore the analysis of integrin a5 and integrin b1 is more informative in this context/

The study was on both kidney and lung of the rats, therefore the tittle does not reflect the content. 

The reviewer is correct. The title was adjusted to include the results of the lungs.

7. Introduction should be revised to include the existing discoveries, shortcomings and the way forward. 

We feel that our introduction gives a nice overview of the current literature and we emphasize that sex differences are not taken into account. Of course we might have missed an important subject and would therefore like to ask the reviewer to be more precise so we can adjust the introduction. 

The aim of the study should be redefined. 

We noticed that our primary and secondary outcome were reversed in the aim and have adjusted this as follows: “Here, we have investigated the effect of sex on renal and pulmonary edema in heterozygous mice with genetically reduced endothelial Tie2 expression, thereby mimicking the suppressive effect of THS on expression of the endothelial Tie2 receptor. In addition, we have investigated whether expression of components of the endothelial angiopoietin/Tie2 system in kidneys and lungs differs between healthy female and male mice.”

The emphasis is on the kidney and the study on both lung and kidney. 

We thank the reviewer for his/her critical reading and have adjusted the introduction accordingly. 

Liver should have been included. 

In our opinion, all vital organs are important as traumatic hemorrhagic shock affects the whole body. We have however focused on the two main affected vital organs, the lung and kidney. We share the opinion that the liver is of interest and in our new study we have included the liver. 

8. The methods are technical but grossly inadequate. Source, age, weight and strain of the mice are not mentioned.

We have added more information regarding the origin of the line to the methods section. In short, mice were bred in the animal facility of the VU University, Amsterdam, the Netherlands. The creation of the line is described in reference Jongman et al. (PMID: 30520765). Mice were included based on age, therefore weight of the mice is only mentioned in the results section as it was not an inclusion criterion. 

Period of acclimatization; cyclical diurnal changes, feeds, period of experimentation, vital parameters and generation of the mice (1st filial or 2nd filial) are not mentioned. 

More information regarding housing, food, water and generation of the mice was added to the methods section. As mice were breeding surplus coming from the animal facility of the VU University, acclimatization was not required as no transport was applied. Additionally, period of experimentation (‘mice were included at the age of 3 to 6 months’) was noted in the methods section. Vital parameters were not measured as mice were only sacrificed.

Euthanasia using isoflurane is controversial. It is not the best drug to use. It causes behavioral changes such as induction of attention deficit, increased anxiety etc. Subheadings under Methods should be rearranged as follows: Experimental set up, Animals and Genotyping, Renal and pulmonary edema formation, Plasma analysis, Renal analysis and study approval. The use of isoflurane and decapitation is a great cruelty to Animal kingdom especially the species of mice. It is quite disheartening. Inhaled isoflurane cause pulmonary edema by attenuating histologic lung injury. I am quite wary of the use of isoflurane/decapitation. Note that isoflurane (5%) can cause death in 60 sec (too long) for humane killing. Respiratory arrest takes longer time.

Termination by decapitation under brief isoflurane inhalation anesthesia is an accepted method in the Netherlands, elsewhere in the EU, and in full compliance with the directive 2010/63/EU. The statement 'Inhaled isoflurane cause pulmonary edema by attenuating histologic lung injury.’ is somewhat confusing. Providing the corresponding reference would have been informative. Generally, beneficial effect of brief isoflurane anesthesia has been reported (PMID: 26068207) and also in combination with the avoidance of stressors by acute brain injections in mice (PMID: 33607166). Even if isoflurane would promote renal or lung edema formation, the mice are killed in an unconscious state immediately (seconds) after inhalation. It is therefore highly unlikely that mice develop pulmonary edema in the few seconds of anesthesia. We have also discussed this issue with our local animal welfare body and there is absolutely no animal welfare concern regarding this procedure.

9. Statistical methods should be revised. All the data presented in bar charts should be revised. Some data generated are quantitative while others are qualitative. The conclusion derived from the study based on the result is quite unfounded. The presented results are not adequately comprehensive. Therefore, I am of opinion that some results should be changed to tabular form, since males and females were used for the study. 

We thank the reviewer for this suggestion. We agree with the reviewer that a large amount of data is presented in figure format. To improve readability of the data we attempted to present a part of it, the PCR data to be specific, in tabular format. Unfortunately, during the execution we came to the conclusion that presenting this data in tabular format did not improve the readability due to the amount of decimals. In our opinion, the readability even decreased when presenting this data in tabular format. We considered presenting the data as normalized to WT males, however this would make it difficult to compare WT females vs Tie2+/- females or Tie2+/- males vs Tie2+/- females. To give an impression of the tables we created, we added tables below. If the reviewer has any other ideas on how to improve the presentation of our data, we are very open to receive any suggestion.

Quantitative data could be analyzed using student test unpaired.

We tested the normality of data distribution using the Shapiro-Wilk test. As data appeared non-normally distributed, it is our opinion that according to good statistical practices, the data should be evaluated with Kruskal-Wallis test.

Specific relevant changes in the gene of each mice should be computed and the differences in the genes could be ranked.

Unfortunately, we do not understand the suggestion of the reviewer. Perhaps he/she could explain us the proposed analysis? 

The Results section requires critical grammatical review. 

We critically read our results section and revised it where necessary.

Functional parameters of kidney should have been measured.

Although the amount of material was limited, as the study was executed on surplus mice coming from the breeding population, we tested several parameters of kidney function. Specifically, we analyzed circulating NGAL and gene expression levels of NGAL and Kidney Injury Marker-1 (KIM-1). Unfortunately, additional analyses such as the evaluation of creatinine or urine expression levels of NGAL appeared unfeasible due to the limited amount of blood or the absence of an urine sample. However NGAL and KIM-1 have been described by many others as indicators of renal injury. 

10. Discussion should be revised. Findings should be discussed with reference to the past findings. 

 We critically reviewed our discussion and revised it were necessary.

11. Conclusion is grossly inadequate and biased. Hence kindly revised the conclusion.

We critically reviewed our conclusion and revised it to fit the presented results and discussion.

 

Reviewer #2: 

We would like to thank the reviewer for putting his/her time and care in providing valuable comments which have significantly improved the manuscript. We answered the comments below in a point-wise manner. Changes and additions in the manuscript based on the provided comments are shown via track changes.

1. Introduction. It is informative with compatible references

We thank the reviewer for this positive feedback.

2. Methods: other than the sample size are small, it is clear and concise. Small sample size may bias the results. 

We agree with the reviewer that the samples size is small. We did however perform a sample size calculation based on preliminary results. 

3. Procedures of methods: clear and concise 

We thank the reviewer for this positive feedback.

4. Statistical analysis: I think with a small number of animals, it is difficult to get statistically significant. 

Based on the samples size calculation we have included 9 mice per group. The sample size calculation is performed on the primary outcome, namely wet/dry weight ratios, and might therefore be not adequate for the secondary outcomes. 

5. The results were presented as figures, which appeared a little bit confused. The other option is to tabulate some of your findings, which be of high values to the readers as well as you impressed them. 

We thank the reviewer for this suggestion. We agree with the reviewer that a large amount of data is presented in figure format. To improve readability of the data we attempted to present a part of it, the PCR data, in tabular format. Unfortunately, during the execution we came to the conclusion that presenting this data in tabular format did not improve the readability due to the amount of decimals. In our opinion, the readability even decreased when presenting this data in tabular format. We considered presenting the data as normalized to WT males, however this would make it difficult to compare WT females vs Tie2+/- females or Tie2+/- males vs Tie2+/- females. To give an impression of the tables we created, we added the tables below.. If the reviewer has any other ideas on how to improve the presentation of our data, we are very open to receive any suggestion.

6. Please, revised your conclusion to be more informative. 

We critically reviewed our conclusion and revised it to fit the presented results and discussion.

 

Reviewer #3: 

We would like to thank the reviewer for putting his/her time and care in providing valuable comments which have significantly improved the manuscript. We answered the comments below in a point-wise manner. Changes and additions in the manuscript based on the provided comments are shown via track changes.

Comments to Author: The role of biological sex as a risk factor for adverse outcomes following major trauma has been widely recognized. Females may confer a protective effect against organ failure, sepsis, and mortality in patients experiencing traumatic hemorrhage. The endothelial angiopoietin/Tie2 system is well recognized for controlling the permeability of endothelial cells. In this study, in heterozygous Tie2 knock-out mice, it appears that the female sex provides protection against renal edema while not exerting the same protective effect against pulmonary edema. This study suggests that there is a sexual dimorphism that provides protection against renal edema in female mice. The study indicated that Tie2 has exhibited potential in mitigating organ edema induced by hemorrhagic shock, specifically in males. However, it is imperative to acknowledge the potential existence of sexual dimorphism in the endothelium, an aspect that has not been investigated in this particular context before. 

Major Comments:

• The authors of the paper have done an impressive job in carrying out their research and presenting their findings. They demonstrated a good level of skill and attention to detail in executing the study and analyzing the data. Additionally, they extensively reviewed the existing literature related to their research topic, providing a comprehensive understanding of the subject matter. 

We would like to thank the reviewer for his/her kind words.

• However, it is important to note that the experimental design employed in the study could have been more carefully chosen. This means that there may be some limitations in how the study was designed. These flaws could potentially impact the reliability of the results obtained. 

We agree with the reviewer that the chosen experimental design may not be the most optimal. We would like to emphasize that the study was carried out using breeding surplus, minimizing the possibilities in choice for experimental design. However, by doing so, animals coming from breeding that are otherwise sacrificed without inclusion in an experiment, are hereby effectively used as data is derived from these animals. This contributes to the reduction of animals and refinement of animal experiments (3R’s). 

• The authors investigated "whether expression of components of the endothelial angiopoietin/Tie2 system in kidneys and lungs differs between healthy female and male mice. The effect of sex on renal and pulmonary injury in heterozygous mice with genetically reduced endothelial Tie2 expression, thereby mimicking the suppressive effect of THS on expression of the endothelial Tie2 receptor." Although the tie-2 deletion has been universal, the authors failed to explain the organ-specific effect of this particular phenotype.

The reviewer addresses an interesting point. Based on the results of the current study, it is difficult to elaborate on the underlying mechanism leading to this organ-specific effect of the heterozygous Tie2 knockout. Although we do not fully understand the mechanism yet, our previous studies also showed an organ-specific effect of targeting Tie2 in a model for hemorrhagic shock (PMID: 28968277). In that specific study, we showed that targeting Tie2 with a Tie2 agonist could reduce pulmonary vascular leakage, but not renal vascular leakage after hemorrhagic shock, even when both organs had reduced Tie2 expression following hemorrhagic shock. Differences in the effectivity of a certain therapy, but also organ-specific differences in response to a heterozygous knock-out could be explained by several mechanisms, but still requires additional research to draw definitive conclusions.

• The discrepancy between gene and plasma expression has not been discussed.

We agree with the reviewer that there is a discrepancy between the expression levels measured in the circulation and on genetic level. Both measurements have advantages and disadvantages. Measuring protein level in plasma is a technique often used in the clinical setting, as plasma is then easily available but organ-specific tissue most often not. However, measurements perform in plasma only give insight in deviations on a circulatory level, not on organ-specific level. As we measured a broad spectrum of genes and proteins in circulation, lung and kidney, it is difficult to include the explanation of the discrepancy in the discussion as this differs per gene and per organ. 

• The abstract stated "estrogens and other endothelial barrier regulators," although only estrogen receptors have been analyzed. While estrogen has been briefly discussed with regards to its role in vasoprotective effects, the postulated selective role of TIE-2 has not been optimally elucidated in this paper. 

First of all, the text in the abstract was adjusted to “estrogen receptors and other endothelial barrier regulators”.

Second, to our knowledge, there are only a few studies discussing an (indirect) link between Tie2 and the possible protective effect of sex. As far as we know, the known literature is included in the discussion. 

• The sample size is relatively small for assessing the claimed hypotheses. 

We agree with the reviewer that the samples size is small. We did however perform a sample size calculation based on preliminary results. 

• Ln 159: Histology has not been presented anywhere in the present study, and the removal of figures has not been justified.

Images of the immunohistochemical analyses were added to the results section (Figure 5C-D). 

Minor Comments:

o Reference [5] are repeated many times. 

We agree with the reviewer that reference [5] is used multiple times. To our knowledge, this is so far one of the few studies describing a link between sexual differences and organ failure in patients following traumatic hemorrhagic shock. If the reviewer has any other ideas, we are open to his/her suggestions. 

o A pharmacological blocker could have a superior outcome, although this has not been utilized in the study and previously utilized by your group 

We fully agree with the reviewer that the addition of an intervention, such as administration of an agonist of antagonist of the angiopoietin/Tie2 system would improve the impact of the study. As the study was carried out with breeding surplus, this was not a possibility in the current set-up. Additionally, the current study solely had as goal to provide insight in basal expression levels of the angiopoietin/Tie2 system and the existing differences between male and female mice. As we see the additional value of interfering in the angiopoietin/Tie2 system, this is part of one of our newly designed studies.

o Ln 81, typo error 

We thank the reviewer for reading our manuscript in a thorough manner. However, we did not observe a typo error in line 81. Perhaps the reviewer could point out which typo error he/she detected.

o The wet/dry weight ratio approach did not show any significant difference in the present study. I wonder if there is a better mechanistic approach to show the investigated phenotype (lung vs. renal edema).

We agree with the reviewer that the measurement of wet/dry weight ratio may not be the most ideal measurement to investigate edema. Ideally, one would like to measure fluid extravasation via determination of FITC-labeled dextrans or Evans Blue extravasations. These techniques have the advantage that only extravasation of a specific molecule size is determined. Unfortunately, with the current set-up we were limited to non-invasive techniques, as we performed the study with breeding surplus, not allowing us to inject any substances. Therefore, we were limited to the determination of wet/dry weight ratio.

o Ln 188: "Circulating levels of angiopoietin-1 (p=0.06) and angiopoietin-2 were higher in females compared to males (Fig 1A-B)" is not entirely supported by the present findings and referenced figures 

In the first section of the results, we describe the differences between wild-type male and female mice. In these mice circulating angiopoietin-1 and angiopoietin-2 were significantly increased in female mice compared to male mice, as shown in figure 1A and 1B (non-striped bars). To emphasize that this part of the results only describes differences between wild-type makes and females, adjusted the sentence to: “In wild-type mice, circulating levels of angiopoietin-1 (p=0.06) and angiopoietin-2 were higher in females compared to males (Fig 2A-B), whereas angiopoietin-2/angiopoietin-1 ratio or soluble Tie2 levels were comparable between males and females (Fig 2C-D).”

o Ln 199 analyzed soluble tie2, I wondered if the cellular lysate of the compared organs might show a better conclusion.

We agree with the reviewer that analyzing cellular lysate would be of high value. Ideally, we would like to measure the phosphorylation status of Tie2 to investigate whether there is a difference in activated Tie2 rather than only measuring total Tie2 or soluble Tie2. Unfortunately, several limitations existed:

- To measure total Tie2 in cellular lysate: the amount of material to do so was not available. One lung/kidney was used for wet/dry weight ratio measurements, where the remaining parts were used for mRNA measurements and IHC stainings.

- To measure pTie2: Until now no specific antibodies are available to determine pTie2 in the kidney or lungs of mice. We recently tested several newly available antibodies, but none of them was specific enough to produce reliable, reproducible results.

o Ln 202 *, **, and *** symbols have not been included in the study. 

We apologize for this inconsistency. The figures are adjusted to match the correct significance indication.

o Not all selected genes are relevant to the proposed mechanism. 

In this study we tested a broad spectrum of genes, directly or indirectly connected to the purpose of the study:

- We analyzed KIM1 and NGAL mRNA levels to discuss the effect of heterozygous Tie2 knockout on renal injury.

- We analyzed angiopoietin-1, angiopoietin-2 and Tie2 to directly measure the effect of Tie2 knockout on the angiopoietin/Tie2 system.

- As downstream targets of the angiopoietin/Tie2 system we measured RhoA and Rac-1.

- Integrin α5 and β1 and VEGFα affect angiopoietin/Tie2 signaling and were therefore added as indirect targets.

- To elaborate on a link between sex and angiopoietin/Tie2 signaling, we investigated estrogen receptor α, estrogen receptor β.

The direct targets of the angiopoietin/Tie2 system have also been visualized in figure 1.

o Protein expression methods other than ELISA might also provide additional insight (e.g. western blotting).

The reviewer is correct. However, due to a limitation in the amount of tissue and the absence of specific antibodies to determine e.g. p-Tie2, we were not able to include western blot analysis in the current study.

o The selected cohort of genes is not fully justified. The molecules included in the analysis were estrogen receptor α, estrogen receptor β, integrin α5, integrin β1, Rac1, RhoA, vascular endothelial growth factor α (VEGFα) (why not other isoforms), kidney injury molecule 1 (KIM1), and neutrophil gelatinase-associated lipocalin (NGAL). 

We feel like this question has been answered above. 

o Estrogen roles have been suggested in many reports but were not sufficiently discussed 

To our knowledge, there is limited data available regarding the link between Tie2 and estrogens. As far as we know, the literature available has been included in the discussion of our manuscript. Future research should elaborate more on the link between Tie2 and estrogen, for example by overexpressing estrogen via estrogen administration, or reducing estrogen levels by for example an ovariectomy. 

o TIE-2 has been proposed as a protective factor, although in vitro studies have been lacking. 

As Tie2 is a flow-responsive gene, and cell culture is mainly static, it is difficult to investigate the protective effects of Tie2 in an in vitro setting. However, there are several studies, performed by our group but also by other groups, that have investigated the protective effects of Tie2 in an in vivo setting. For example, administration of Tie2 agonist vasculotide improves microcirculatory perfusion and reduces microvascular leakage in animals following hemorrhagic shock or cardiopulmonary bypass (PMID: 28968277, PMID: 30336848). Also, in mice with acute kidney injury, enhancing Tie2 phosphorylation could improve renal perfusion and reduce renal microvascular leakage (PMID: 26911791).

 

Reviewer #4: 

Good work. The paper is well written and discussed found that Female sex seems to protect against renal, but not pulmonary edema in heterozygous Tie2 knock-out mice. This could not be explained by sex dimorphism in the endothelial angiopoietin/Tie2 system. 

We would like to thank the reviewer for putting his/her time in reading our manuscript critically. We also would like to thank the reviewer for the positive feedback. 

 

Reviewer #5: 

We would like to thank the reviewer for putting his/her time and care in providing valuable comments which have significantly improved the manuscript. We answered the comments below in a point-wise manner. Changes and additions in the manuscript based on the provided comments are shown via track changes.

Leeuwen and colleagues have investigated the effect of gender on edema formation in mice with partial deletion of Tie2.

Major concerns:

1- The authors have to justify why they did not use mice with full Tie2. 

We agree with the reviewer that it would be very interesting to investigate the effect of a full Tie2 knock-out. Unfortunately, these mice are lethal as the expression of Tie2 is essential in the development of, amongst others, the vascular system. Therefore, only a heterozygous knock-out is viable.

In addition, please include the genotyping bands of three random mice in each group (in the supp material). 

We thank the reviewer for this suggestion. We included a supplementary figure to show a typical example of genotyping results in both males and females. (supplementary figure 1)

2- The authors relied on the w/d ratio and certain markers to conclude the edema formation. This needs further investigations such as, the albumin contents of the kidney tissues and BAL fluid as well as IHC analysis of the kidney and lung tissue to evaluate the thickness of interalveolar septa or the tubules in the kidney tissues. 

Indeed additional analyses would be interesting to elaborate more on the level of organ injury. In the current study we were limited to the usage of non-invasive techniques, as the study was carried out on breeding surplus. Therefore, BAL fluid analysis was not feasible. We did, however, perform IHC analysis of kidney and lung tissue. Results of these analysis are visualized in figure 5C and 5D.

3- Tie2 receptor known to be extensively regulated on post-translational level, thus, the tissue levels of Tie2 need to be analysed (by WB or ELISA).

The reviewer addresses an important point. It would, indeed, be of high additional value to include data regarding the phosphorylation status of Tie2 in tissue. Unfortunately, due to the limited amount of tissue available and the absence of specific antibodies, this was not possible in the current study. We recently tested several newly available antibodies for p-Tie2, but none of them appeared specific enough to generate reliable, reproducible data.

Minor concerns:

1- please label in each figure the corresponding tissue (kidney, pulmonary...). 

The figures were adjusted as suggested.

2- please show the IHC data (stated in the results as data not shown).

We agree with the reviewer that visualization of the IHC data improves the interpretation of the results for the reader. We therefore included a typical example of each group in Figure 5C and 5D..

3- The authors can design a scheme to describe the proposed mechanism.

With the current study we aimed to provide a first insight in basal differences between male and female mice and the difference in effect of heterozygous Tie2 knockout between males and females. Unfortunately, the literature that describes a possible link between sex and Tie2 signaling is very limited and also based on the results of the current study it is difficult to propose a possible underlying mechanism for the found differences. In the current stage of this research topic it is therefore difficult to propose a specific mechanism. 

 

Reviewer #6: 

We would like to thank the reviewer for putting his/her time and care in providing valuable comments which have significantly improved the manuscript. We answered the comments below in a point-wise manner. Changes and additions in the manuscript based on the provided comments are shown via track changes.

Thank you for nice work and research. 

We thank the reviewer for his/her kind words.

I have couple of recommendations:

- Line 72-82 Add a simplified and summarized figure about the angiopoietin/Tie2 signaling pathway 

We thank the reviewer for this suggestion. We added a schematic overview of angiopoietin/Tie2 signaling to the introduction (Fig. 1).

- Line 305-307 rewrite the two sentences regarding the pulmonary and renal gene expression to avoid duplication

We thank the reviewer for reading the manuscript in such a detailed manner and changed the sentences to avoid duplication as follows: “Renal gene expression of integrin α5, integrin β1, VEGFα, RhoA and Rac1 was not affected by sex nor differed between Tie2+/- and Tie2+/+ mice (Fig 6C-G). In lungs, RhoA gene expression was higher in female Tie2+/- mice compared to female Tie2+/+ mice (Fig 7F), whereas gene expression of integrin α5, integrin β1, VEGFα, and Rac1 was not affected by sex nor differed between Tie2+/- and Tie2+/+ mice (Fig 7C-E,G).”

- All Figs (1B, 1D, 2A-C, 4A, 4C, 4D, 5B, 5E, & 6F) need to be revised and fix typo of (1) 

We apologize for this inconsistency. The figures are adjusted to match the correct significance indication.

---

## [Decision Letter · Decision Letter 1]

18 Oct 2023

Female sex protects against renal edema, but not lung edema, in mice with partial deletion of the endothelial barrier regulator Tie2 compared to male sex

PONE-D-23-11857R1

Dear Dr. van den Brom,

We’re pleased to inform you that your manuscript has been judged scientifically suitable for publication and will be formally accepted for publication once it meets all outstanding technical requirements.

Kind regards,

Keiko Hosohata, Ph.D.

Academic Editor

PLOS ONE

Reviewers' comments:

Reviewer's Responses to Questions

**Comments to the Author**

1. If the authors have adequately addressed your comments raised in a previous round of review and you feel that this manuscript is now acceptable for publication, you may indicate that here to bypass the “Comments to the Author” section, enter your conflict of interest statement in the “Confidential to Editor” section, and submit your "Accept" recommendation.

Reviewer #7: All comments have been addressed

Reviewer #8: All comments have been addressed

2. Is the manuscript technically sound, and do the data support the conclusions?

Reviewer #7: Yes

Reviewer #8: Yes

3. Has the statistical analysis been performed appropriately and rigorously? 

Reviewer #7: Yes

Reviewer #8: Yes

4. Have the authors made all data underlying the findings in their manuscript fully available?

Reviewer #7: Yes

Reviewer #8: Yes

5. Is the manuscript presented in an intelligible fashion and written in standard English?

Reviewer #7: Yes

Reviewer #8: Yes

6. Review Comments to the Author

Reviewer #7: (No Response)

Reviewer #8: This is interesting paper. The authors answered questions. Authors shown sex dimorphism in the endothelial angiopoietin/Tie2 system.

7. PLOS authors have the option to publish the peer review history of their article (what does this mean?). If published, this will include your full peer review and any attached files.

Reviewer #7: No

Reviewer #8: No

---

## [Editor Report · Acceptance letter]

7 Nov 2023

PONE-D-23-11857R1 

Female sex protects against renal edema, but not lung edema, in mice with partial deletion of the endothelial barrier regulator Tie2 compared to male sex 

Dear Dr. van den Brom:

I'm pleased to inform you that your manuscript has been deemed suitable for publication in PLOS ONE. Congratulations! Your manuscript is now with our production department. 

Kind regards, 

on behalf of

Dr Keiko Hosohata 

Academic Editor

PLOS ONE